# A Case of Focal Segmental Glomerulosclerosis in a Young Girl with a Very Low Birth Weight

**Yasuyo Kashiwagi \*, Kazushi Agata, Gaku Yamanaka** 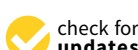 **and Hisashi Kawashima**

Department of Pediatrics and Adolescent Medicine, Tokyo Medical University, Tokyo 160-0023, Japan; whatwouldjesusdo721@yahoo.co.jp (K.A.); gaku@tokyo-med.ac.jp (G.Y.); hisashi@tokyo-med.ac.jp (H.K.)
\* Correspondence: hoyohoyo@tokyo-med.ac.jp; Tel.: +81-3-3342-6111; Fax: +81-3-3344-0643

**Abstract:** In Japan, the prevalence of low birth weight (LBW) has been estimated to be approximately 10%, which is the highest among developed countries. This high prevalence might affect the occurrence of LBW-associated diseases in the adult population of Japan. LBW has been recognized as a contributing factor to post-adaptive focal segmental glomerulosclerosis (FSGS) in adulthood; however, few reports to date have evaluated the clinical and pathological characteristics of post-adaptive FSGS. A 13-year-old girl was referred to our hospital owing to mild proteinuria, which was detected at a school urinary screening. She was born at a gestational age of 23 weeks, with a very LBW of 630 g. Dipstick urinalysis revealed grade (2+) proteinuria. Her serum creatinine level was 1.02 mg/dL, and she was diagnosed as having stage 2 chronic kidney disease (CKD). Her serum uric acid level was 7 mg/dL. Furthermore, her mother and 16-year old brother had hyperuricemia. A percutaneous renal biopsy leads to a diagnosis of FSGS. After 3 years of treatment with an angiotensin receptor blocker, her proteinuria decreased. However, her serum creatinine level was 1.07 mg/dL, and she still had stage 2 CKD. We considered that in this patient, the first hit was her LBW, and the second hit was hyperuricemia. The second hit might be associated with the development of CKD. The birth history of patients is not usually confirmed by nephrologists. Our case demonstrates that obtaining information regarding the preterm birth and LBW of patients is important in the diagnosis of noncommunicable diseases because school urinary screening is not routinely performed in countries other than Japan.

**Keywords:** chronic kidney disease; low birth weight; focal segmental glomerulosclerosis; two-hit theory

## 1. Introduction

Preterm birth rates have recently increased worldwide, and approximately 13 million infants are born preterm each year [1]. Clinical and experimental lines of evidence have demonstrated that preterm and low birth weight (LBW) infants are at risk of hypertension and cardiovascular and renal dysfunction in adulthood [2]. This is the concept of the developmental origins of health and disease, which play a potential role in determining the susceptibility to developing noncommunicable diseases (such as cardiovascular disease and chronic kidney disease (CKD) in adulthood [3]. In Japan, the prevalence of LBW has been estimated to be approximately 10%, which is the highest in developed countries. Therefore, this high prevalence might affect the occurrence of LBW-associated diseases in the adult population of Japan [4]. LBW has been recognized as a contributing factor to the development of post-adaptive focal segmental glomerulosclerosis (FSGS) in adulthood [5]. However, few studies to date have evaluated the clinical and pathological characteristics of post-adaptive FSGS.

## 2. Case Report

A 13-year-old girl was referred to our hospital due to mild proteinuria which was detected at a school urinary screening. No urinary abnormalities had been detected previously. She was born at a gestational age of 23 weeks and 6 days because of maternal fever, with a very LBW of 630 g, equivalent in size to 23 weeks of pregnancy. She had chronic lung disease and premature infant retinitis. She required O$_2$ until she was 1 year of age; however, her growth and development had reached the normal range by that time. Apart from retinopathy of prematurity, she had been generally healthy, both physically and mentally.

Physical examination showed the following: height 154 cm, weight 50 kg, body mass index 21.1, and blood pressure 115/73 mmHg. Dipstick urinalysis revealed grade (2+) proteinuria. Her serum creatinine level was 1.02 mg/dL, and she was diagnosed as having stage 2 CKD based on reference serum creatinine levels of Japanese male and female children aged 12 to 15 years. Her serum uric acid level was 7 mg/dL (normal level was less than 4.6 mg/dL based on reference serum levels of female children aged 12 to 14 years), she had hyperuricemia [6]. Her mother and her 16-years old brother had hyperuricemia, too.

A percutaneous renal biopsy demonstrated that 1 out of 8 glomeruli had segmental sclerosis with adhesion to the Bowman's capsule (black arrow), and white arrow indicated partial focal interstitial fibrosis (Figure 1). The mean diameter of the glomeruli was 348.23 μm, which was much larger than that of normal glomeruli (168 ± 12 μm) [7]. Immunofluorescence analyses (IgG, IgA, IgM, C3, C1q and C4) were all negative. These biopsy findings supported a diagnosis of FSGS.

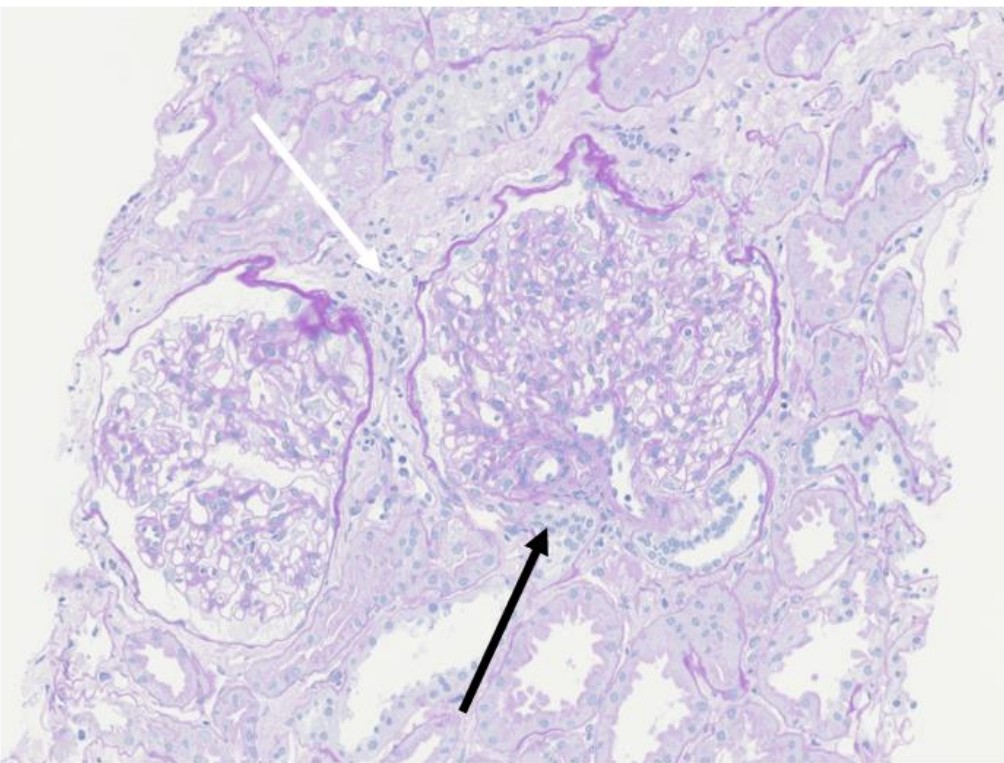

**Figure 1.** The mean diameter of the glomeruli of this patient (348.23 μm) was much larger than that of normal glomeruli (168 ± 12 μm). Periodic Acid Schiff staining (magnification: 200×). White arrow: partial focal interstitial fibrosis. Black arrow: segmental sclerosis with adhesion to the Bowman's capsule.

After 3 years of treatment with an angiotensin receptor blocker (ARB), her proteinuria decreased. However, her serum creatinine level was 1.07 mg/dL, and she still had stage 2 CKD.

## 3. Discussion

In Japan, birth weights have decreased over the years, resulting in decreased renal function in some healthy Japanese adolescents [8]. Birth weight linearly correlates with the total number of functional nephrons. In individuals born with a LBW, a congenital deficit of nephrons results in compensatory glomerular hyperfiltration, hypertrophy, and hypertension. The degree of susceptibility to the development of CKD is known to have a sex difference [9], and it has been demonstrated that renal function decreases more slowly in women than in men, and the effect of LBW on CKD is weaker in women [10]. Other than sex, several factors affect the development of CKD, such as race, obesity, hypertension, etc. Das et al. demonstrated that at least one-third of people with diabetes are born with a LBW and develop, and hence this risk is of great importance [11].

Even if an individual develops proteinuria, there may not be any abnormalities evident on renal biopsy. Hayashi et al. reported that 5 children born with very LBWs were diagnosed as having proteinuria, but their renal biopsies showed no findings of mesangial proliferation or glomerular sclerosis, although the mean diameter of their glomeruli was larger than that of normal glomeruli [12]. A second-hit exposure might be associated with the development of CKD. In the present case, the first hit might be the LBW, and the second-hit exposure might be hyperuricemia. Plasma uric acid levels are increased because of a decrease in glomerular filtration rate. The patient continued to show an increased serum creatinine level and might hence require urate-lowering therapies in the future. Clinical studies have demonstrated that urate-lowering therapies might help to prevent and delay the worsening of CKD [13].

ARBs reduce glomerular hypertension and have been reported to decrease proteinuria. In our present patient, her proteinuria decreased after 3 years of treatment with ARB. The birth history of patients is not usually confirmed by nephrologists. Obtaining information regarding the preterm birth and LBW of patients is important in the diagnosis of noncommunicable diseases because urinary screening is not routinely performed in schools in countries other than Japan.

**Author Contributions:** Y.K. wrote the manuscript; Y.K., K.A. and G.Y. contributed to the conception and design of the manuscript; H.K. critically reviewed the manuscript and supervised the whole study process. All authors have read and agreed to the published version of the manuscript.

**Funding:** This research received no external funding.

**Institutional Review Board Statement:** Not applicable.

**Informed Consent Statement:** Written informed consent has been obtained from the patient to publish this paper.

**Data Availability Statement:** Not applicable.

**Conflicts of Interest:** The authors declare no conflict of interest.

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
