# Peer review of "A Case of Focal Segmental Glomerulosclerosis in a Young Girl with a Very Low Birth Weight"

_pediatrrep, doi:10.3390/pediatric14020022_

Round 1

Reviewer 1 Report

In this paper, the authors describe the clinical case of focal segmental glomerulosclerosis in a young girl. The girl had proteinuria and was diagnosed with stage 2 chronic kidney disease. The authors highlighted that girl was born with low birth weight and associated this with FSGS in adulthood. This manuscript demonstrated that relevant patient birth information could be significant in diagnosing and predicting kidney disease. The authors provide important data for evaluating the clinical and pathological characteristics of FSGS.

Only a few issues need to be revised:

1. Line 45 - I would suggest adding more recent references or avoiding the word "Recently." I think 13 years old reference should not be considered as "recent" in the context of this paper because in your references oldest paper is 20 years old, which is not so far from 13 years old.

2. Line 66 has reference to Fig.1. The figure doesn't have pointers (like a - Bowman's capsule adhesion), highlighting evidence of partial focal interstitial fibrosis. I would suggest a more comprehensive description of the figure. 

3. Line 15 and 44 - I suggest rephrasing "...prevalence might affect the prevalence..."  doesn't look good. Consider using a synonym in its place.

Author Response

Reviewer 1

Thank you for your comments.

1: Line 45- I avoided “Recently”.

2: Line 64-66 I made more comprehensive description of the figure.

A percutaneous renal biopsy demonstrated that 1 glomerulus had segmental sclerosis with adhesion to the Bowman’s capsule, and partial focal interstitial fibrosis was also observed (Figure 1).

âž¡A percutaneous renal biopsy demonstrated that 1 out of 8 glomerulus had segmental sclerosis with adhesion to the Bowman’s capsule (black arrow), and white arrow indicated partial focal interstitial fibrosis (Figure 1).

3: Line 15 and 43-44

I revised this sentence.

Therefore, this high prevalence might affect the prevalence of LBW-associated diseases in the adult population of Japan.

âž¡Therefore, this high prevalence might affect the occurrence of LBW-associated diseases in the adult population of Japan.

Reviewer 2 Report

Thank you for allowing me to review this work by Kashiwagi et al describing a case report of focal segmental glomerulosclerosis in a young girl 2 with a very low birth weight. 

Please refer to my comments to the editors.

Author Response

Reviewer2

Thank you for your comments. Please see the attachment.

Reviewer 3 Report

Please provide information in the text if this patient was born with  fetal growth restriction ( birth wt < 10 % of expected for geatational age ) .It is important to mention that the embryonic  development of the kidneys is by branching morphogenesis and nephrons are added centrifugally till about 36 weeks of gestation .Those born prematurely ,LBW and with fetal growth restriction have low nephron endowment   which in turn  results in  hyperfiltration and later development of proteinuria, hypertension and CKD .

Author Response

Reviewer3

Thank you for your comments.

I added the information about fetal growth in the line 52-53.

She was born at a gestational age of 23 weeks, with a very LBW of 630 g, and had chronic lung disease and premature infant retinitis.

âž¡She was born at a gestational age of 23 weeks and 6 days because of maternal fever, with a very LBW of 630 g, equivalent in size to 23 weeks of pregnancy. She had chronic lung disease and premature infant retinitis.

Round 2

Reviewer 2 Report

With further review and in support of my fellow colleagues' reviewers, I am happy to support the decision to accept this manuscript in your journal. I appreciate the healthy discussions and the efforts the authors put in the manuscript.   Best regards, Omar